# Investigation of Mechanical and Corrosion Properties of New Mg-Zn-Ga Amorphous Alloys for Biomedical Applications

**DOI:** 10.3390/jfb15090275

**Published:** 2024-09-20

**Authors:** Viacheslav E. Bazhenov, Mikhail V. Gorobinskiy, Andrey I. Bazlov, Vasiliy A. Bautin, Andrey V. Koltygin, Alexander A. Komissarov, Denis V. Ten, Anna V. Li, Alexey Yu. Drobyshev, Yoongu Kang, In-Ho Jung, Kwang Seon Shin

**Affiliations:** 1Casting Department, National University of Science and Technology “MISiS”, Leninskiy pr. 4, 119049 Moscow, Russia; mgorobinskiy@mail.ru (M.V.G.); misistlp@mail.ru (A.V.K.); 2Laboratory of Advanced Green Materials, National University of Science and Technology “MISiS”, Leninskiy pr. 4, 119049 Moscow, Russia; bazlov@misis.ru; 3Department of Metallurgy Steel, New Production Technologies and Protection of Metals, National University of Science and Technology “MISiS”, Leninskiy pr. 4, 119049 Moscow, Russia; bautin@list.ru; 4Laboratory of Hybrid Nanostructured Materials, National University of Science and Technology “MISiS”, Leninskiy pr. 4, 119049 Moscow, Russia; komissarov.alex@gmail.com (A.A.K.); teden92@yandex.ru (D.V.T.); anna23-95@mail.ru (A.V.L.); 5Laboratory of Medical Bioresorption and Bioresistance, Russian University of Medicine, Dolgorukovskaya 4, 127473 Moscow, Russia; lab_bioresorption@mail.ru (A.Y.D.); ksshin@snu.ac.kr (K.S.S.); 6High Temperature Thermo-Chemistry Laboratory, Seoul National University—Research Institute of Advanced Materials, 1 Gwanak-ro, Gwanak-gu, Seoul 08826, Republic of Korea; yoongukang0209@snu.ac.kr (Y.K.); in-ho.jung@snu.ac.kr (I.-H.J.); 7Magnesium Technology Innovation Center, Department of Materials Science and Engineering, Seoul National University, 1 Gwanak-ro, Gwanak-gu, Seoul 08826, Republic of Korea

**Keywords:** biomaterials, corrosion properties, gallium, magnesium alloys, mechanical properties, metallic glass

## Abstract

Magnesium alloys are considered as promising materials for use as biodegradable implants due to their biocompatibility and similarity to human bone properties. However, their high corrosion rate in bodily fluids limits their use. To address this issue, amorphization can be used to inhibit microgalvanic corrosion and increase corrosion resistance. The Mg-Zn-Ga metallic glass system was investigated in this study, which shows potential for improving the corrosion resistance of magnesium alloys for biodegradable implants. According to clinical tests, it has been demonstrated that Ga ions are effective in the regeneration of bone tissue. The microstructure, phase composition, and phase transition temperatures of sixteen Mg-Zn-Ga alloys were analyzed. In addition, a liquidus projection of the Mg-Zn-Ga system was constructed and validated through the thermodynamic calculations based on the CALPHAD-type database. Furthermore, amorphous ribbons were prepared by rapid solidification of the melt for prospective alloys. XRD and DSC analysis indicate that the alloys with the most potential possess an amorphous structure. The ribbons exhibit an ultimate tensile strength of up to 524 MPa and a low corrosion rate of 0.1–0.3 mm/year in Hanks’ solution. Therefore, it appears that Mg-Zn-Ga metallic glass alloys could be suitable for biodegradable applications.

## 1. Introduction

Millions of people suffer from bone fractures caused by age-related diseases such as osteoporosis and accidents. In these medical cases, permanent metallic implants are often used. However, it has been observed that up to 40% of these implants are symptomatically removed through another surgical intervention [1]. This is due to several disadvantages, including temperature and tactile sensitivity, stress shielding, and detached titanium particles, which can lead to implant-induced inflammation and increased healthcare costs [2]. Magnesium biodegradable alloys are currently being used as commercial fixation devices in orthopedic practices worldwide. Studies have shown that these devices perform equally as well as Ti permanent implants [3,4]. Biodegradable fixation devices that gradually dissolve as the healing process progresses are considered an ideal solution for osteosynthesis [5]. Magnesium alloys are considered a promising option for temporary implants due to their good biocompatibility, high mechanical properties, and acceptable corrosion rate in both in vitro and in vivo corrosion tests. Furthermore, they closely match human bone density and Young’s modulus [6,7,8,9,10]. Previous experimental studies have also demonstrated the positive effects of Mg-based implants on bone metabolism, vascular formation, neuroregeneration, and immunity [11].

While Mg-based alloys are known for their good biodegradability, their degradation rate can be notably rapid due to various factors such as the microgalvanic and interfacial corrosion caused by the presence of secondary phase precipitates, dislocations, grain boundaries, and composition segregation in the alloy structure. As a result, in some cases, the degradation time may be insufficient for the completion of the bone-healing process. Moreover, the release of gaseous hydrogen during the corrosion process can result in the formation of gas gangrene, which may accelerate the degradation of implant integrity and impede the proper connectivity of osteocytes to the implant surface. This could potentially impact bone tissue healing [2,12]. It is well established that Mg is an essential element for human metabolism. However, an overdose can occur due to the extremely high rate of Mg ions release, which can lead to respiratory distress, hypotension, muscular paralysis, and potentially cardiac arrest [13].

Bulk metallic glasses (BMGs) have an amorphous structure that can mitigate the rapid degradation of Mg alloy due to alloy compositional homogeneity at the micro- and macroscale, absence of secondary phase precipitates, and grain boundaries [14]. The Mg-Zn-Ca system is particularly popular among magnesium BMGs due to its good biocompatibility and mechanical properties with potential for medical devices [15,16,17,18,19,20,21]. Previous studies have investigated the quaternary alloys of the Mg-Zn-Ca-X system, where X represents Ag [22,23], Sr [24,25], Mn [26], Li [27], and Ga [28]. Additionally, other systems involving the addition of Cu, Y, Yb, and Ag to Mg have also been studied, albeit to a lesser extent [16,29,30].

As previously mentioned, several studies have shown that Mg has a positive impact on bone regeneration [5,11,15]. However, it is worth noting that other alloying elements can also effectively facilitate the bone repair functions of implant materials. For instance, previous studies have shown that gallium has been effective in treating osteoporosis [31], hypercalcemia [32,33,34], Paget’s disease [35,36], and multiple myeloma [37], and it has an anti-osteoclastic effect [38,39,40]. Additionally, gallium has been identified as a bone resorption inhibitor [41,42]. The hydroxyapatite coating containing Ga on Gription™ implants was found to have doubled the bone growth rate approved in vivo [43].

Various methods, such as equal channel angular pressing (ECAP) [44,45], hot extrusion [46,47], drawing [48], rotary forging [48], and rolling [49], have been used in previous studies to investigate the structure and properties of Mg-Zn-Ga alloys. These studies have found that the alloys exhibit high strength and corrosion resistance in vitro. The use of Mg-Zn-Ga alloys for biodegradable implants may have potential benefits for the bone healing process due to the release of Ga ions during degradation, which could be advantageous when compared to other conventional magnesium alloys.

According to Zai et al. [28], the addition of 1 at.% Ga has been shown to increase the glass-forming ability (GFA) of Mg-Zn-Ca metallic glass. The study also found that Ga addition can help form a passive film on the metallic glass substrate and improve its corrosion resistance as demonstrated by the results of electrochemical tests and immersion tests [28].

Other Ga-containing metallic glasses, such as Mg-Ga, Mg-Ca-Ga, Mg-Ga-Al, Zr-Ga-Ni, and Fe-(Al,Ga)-B, have also been investigated [50,51,52,53,54]. It has been widely acknowledged that compositions near the eutectic point [55] exhibit higher glass-forming ability (GFA). In the Mg-Zn-Ga system, binary and ternary eutectic transitions have been observed in regions close to the Mg corner [45,46,56,57]. Metallic glasses have been obtained previously in the composition range of 0 < x < 0.2 for Mg_0.7_Zn_0.3−x_Ga_x_ [58]. The objective of this study was to analyze the Mg-Zn-Ga system and select the most appropriate alloys for the preparation of amorphous ribbons and subsequent analysis of their microstructure, mechanical properties, and corrosion characteristics. This study presents an initial evaluation of the properties of Mg-Zn-Ga amorphous ribbons for potential use in orthopedic implants.

## 2. Materials and Methods

### 2.1. Alloys Preparation for Preliminary Investigation

The high-purity bulk metals magnesium (99.95 wt.% purity; Solikamsk Magnesium Plant “SOMZ”, Solikamsk, Russia), zinc (99.995 wt.%; Ural Mining and Metallurgical Company “UGMK”, Verkhnyaya Pyshma, Russia), and gallium (99.9999 wt.%; Girmet Ltd., Moscow, Russia) were employed in the preparation of alloys. The melting of the alloys was conducted in a PT90/13 (LAC, Židlochovice, Czech Republic) resistance furnace with a steel crucible. In order to prevent oxidation of the melt during the melting process, carnallite flux was utilized. The ingots, with a 50 mm diameter and a 20 mm height, were poured into a preheated steel permanent mold. A total of 16 alloys with varying Zn and Ga contents were prepared, and their compositions are listed in Table 1. For the analysis of the alloys’ chemical compositions, energy-dispersive X-ray spectroscopy (EDS) was used. The EDS was employed on the alloys’ metallographic sections for three repetitions on an area of 1 × 1 mm^2^ with an accuracy of 0.1 wt.%.

The liquidus projection and primary crystalline phase regions of the Mg-Zn-Ga system, as well as the equilibrium solidification pathways of alloys, were calculated using the FactSage software (www.factsage.com (accessed on 1 August 2024); Thermfact, Montreal, Canada and GTT-Technologies, Aachen, Germany) containing the database developed based on the Calculation of Phase Diagram (CALPHAD) methodology. The thermodynamic database FTlite (version 8.3) was employed in the present calculations.

The temperature during the cooling process of the alloys was recorded using a BTM-4208SD 12-channel temperature recorder (Lutron, Taipei, Taiwan). A K-type thermocouple was installed in the mold before pouring. A scanning electron microscope (Vega SBH3, Tescan, Brno, Czech Republic) with an energy-dispersive X-ray spectrometer (Oxford, UK) was employed to analyze the microstructure of alloys and the elemental content of the phases. Additionally, X-ray diffractometry (XRD) was used to determine the phase composition of #1, #2, #10, and #15 alloys. The measurements were taken with a D8 ADVANCE diffractometer (Bruker, Billerica, USA) under monochromatic Cu Kα radiation.

### 2.2. Alloys Preparation for Melt Spinning and Its Analysis

After conducting a preliminary investigation, a selection was made of the #4, #8, #9, and #13 perspective alloys, and a different melting method was used for master alloys preparation. The alloys were prepared using a resistance furnace with a graphite crucible and cap without any melt protection due to the alloys’ low melting point (<400 °C). The Zn-Ga melt was first prepared, and then the Mg was dissolved at 450 °C. Finally, the crucible with the melt was withdrawn from the furnace and allowed to air-cool.

Ribbon samples were prepared using a rapid solidification process on a single copper wheel in a Vacuum Melt Spinner DX-II (Dexing Magnet tech Company, Xiamen, China) with a silica nozzle in an argon atmosphere. The experimental procedure involved the use of two tangential velocities, namely 12 and 25 m/s. The thickness of the ribbons was approximately 50 µm and 70 µm for velocities of 12 and 25 m/s, respectively, while the width of the ribbons was approximately 1.3 mm and 1.9 mm for velocities of 12 and 25 m/s, respectively.

The solidification range of master alloys and the crystallization temperatures of amorphous ribbons were measured using differential scanning calorimetry (DSC) with Chip-DSC 100 calorimeter (Linseis, Selb, Germany). The measurements were conducted under an Ar gas flow at a heating and cooling rate of 20 °C/min in Al crucibles. The microstructures of the ribbons were also examined using X-ray diffractometry.

### 2.3. Mechanical Properties

The microhardness of the ribbons was measured using an automated UH250 (Wilson Wolpert, Leinfelden-Echterdingen, Germany) microhardness tester. The ribbon was loaded with a force of 25 N for a duration of 15 s.

The mechanical properties of the amorphous ribbons were investigated using a 5966 universal testing machine (Instron, Norwood, MA, USA) with tensile tests performed at a crosshair speed of 0.2 mm/min. For each alloy and tangential velocity, approximately 20 to 25 ribbons were tested.

### 2.4. Corrosion Test

For the in vitro immersion corrosion tests, the ribbons were cleaned in ethanol using an ultrasonic bath. To ensure consistency, a length in the range of 110–250 mm was chosen for each ribbon, taking into account their different thickness and width, to achieve the same surface area of approximately 6 cm^2^. The corrosion testing procedure was conducted for a total duration of 192 h in Hanks’ solution (PanEco, Moscow, Russia). The temperature of the corrosion medium was maintained at a fixed value of 37 degrees Celsius by means of a VT18 thermostat (Termex, Tomsk, Russia). The ratio of solution to specimen surface area was approximately 65 mL/cm^2^. In accordance with the ASTM standard [59], the average corrosion rate (CR) in mm/year has been calculated. The weight loss of the specimens has been determined by measuring the volume of H₂ produced during the corrosion test by specimen and collected by burette. For the calculation of the corrosion rate, the assumption was made that 1 mL of H₂ was proportional to 1 mg of alloy weight loss [44,60,61]. To prevent the amorphous ribbon from floating due to released hydrogen, a polyurethane foam insert was installed into the glass with corrosion media. The pH of the corrosion media was determined during the immersion corrosion test using a HI83141 pH meter (Hanna Instruments, Woonsocket, RI, USA). The analysis was conducted on three to five samples for each alloy composition and rotational frequency.

The potentiodynamic curves of amorphous ribbons in Hanks’ solution were obtained using an IPC Pro MF potentiostat/galvanostat system (Volta, St. Petersburg, Russia). The temperature of the media was maintained at 37 °C using a thermostat. Segments of the amorphous ribbon, each measuring 55 mm in length, were cut. The central portion of the ribbon was insulated at the waterline with 10 mm thick 3M™ Extreme Sealing Tape 4412N. During the measurement process, a ribbon served as the working electrode, with an exposure area of 0.3 ± 0.1 cm^2^, along with a silver/silver chloride (Ag/AgCl) reference electrode and a platinum auxiliary electrode. Prior to the electrochemical analysis, the ribbons were subjected to a short-term immersion in an aqueous solution of 0.3 wt.% HNO₃, which was followed by a rinse in distilled water. Potentiodynamic polarization was conducted, commencing from the cathode region of −1700 mV and progressing to the anode region of +400 mV, with a scan rate of 1 mV/s. To enhance the reliability of the results, five to eight samples were utilized for each measurement. The electrochemical process parameters, including the corrosion potential (*E_corr_*), corrosion current density (*i_corr_*), and anodic and cathodic Tafel slopes, were determined by extrapolating the Tafel region from the polarization curves. The corrosion current densities were employed in calculating the corrosion rate of the ribbons in accordance with the ASTM standard [62].

## 3. Results

### 3.1. Analysis of Solidification and Microstructure of Mg-Zn-Ga Alloys

The equilibrium solidification paths and liquidus and solidus temperatures of all alloys are calculated and presented in Table 1. Figure 1 displays the cooling curves of the Mg-Zn-Ga alloys listed in Table 1. According to the cooling curves, it appears that the solidification process for alloys #1–#13 was completed at a temperature of 312 °C. The binary eutectic transitions observed in the alloys near the Mg corner are consistent with the binary Mg-Ga and Mg-Zn phase diagrams [56,57]. According to the CALPHAD calculations of alloy solidification pathways, it has been observed that for most of the alloys (#1–#13 except #11), solidification concludes with the L→α-Mg + Mg_12_Zn_13_ + Mg_5_Ga_2_ ternary eutectic transition, which takes place at 295 °C as indicated in Table 1 (#11 is finishing at 298 °C). It is worth noting that in previous research [45], the ternary eutectic transition temperature was found to be 307 °C, which is consistent with the present thermal analysis.

The solidification of alloys #14 and #15 appears to end at a higher temperature, which is in agreement with the results of CALPHAD calculations. It is worth noting that the L→MgZn_2_ + MgGa + Mg_2_Ga transition occurs at 317 °C in #14 and #15 alloys according to the calculations. Alloy #16 showed the significant difference between the experimental data and prediction by CALPHAD calculations due to unknown reasons.

Figure 2 shows the as-cast microstructures of all alloys. First of all, the final microstructures show clearly two different groups of alloys: the first group is alloys #1–#10, and the second group is alloys #11–#16. As shown in Table 1, the primary crystalline phase of alloys #1–#8 is calculated to be α-Mg. Alloys #9 and 10 have Mg_5_Ga_2_ as a primary phase, but a large amount of α-Mg can be formed as a second phase and continued until the final eutectic reaction. Therefore, the first group of alloys show similar microstructures. According to the calculations, the second group of alloys (#11–#16) all have primary and secondary crystalline phases other than α-Mg. Most of the alloys have 0 α-Mg or a very small amount of α-Mg in the final equilibrium solidification stage. Although a small amount of α-Mg is still observed in several alloys, the amount of α-Mg is much smaller than for the first group of alloys.

To distinguish the phases in the alloy structure, XRD analysis was carried out on several alloys. The results for alloys #1, #2 and #10, which display the α-Mg, Mg_7_Zn_3_ and Mg_5_Ga_2_ phases, are shown in Figure 3. The alloys have been confirmed to be in the region of the ternary eutectic transition. However, there seems to be a discrepancy between the expected composition of the Zn-rich phase, which should be Mg_12_Zn_13_ according to the calculation results, and the actual composition observed through XRD, which shows the presence of Mg_7_Zn_3_. Further analysis through EDS reveals that the dark phase in the alloys is α-Mg. Additionally, the bright zone in alloys #1–#10 is a mixture of Mg_7_Zn_3_ and Mg_5_Ga_2_ phases, which have almost the same contrast in the BSE regime. It should be noted that (as shown in calculated liquidus projection in Figure 4) Mg_12_Zn_13_ has a very narrow primary crystalline phase region and often does not readily form during solidification; instead, the Mg_2_Zn phase usually forms. It should be noted that the Mg_2_Zn phase is often recognized as Mg_7_Zn_3_. Therefore, the calculated results from the CALPHAD database seem to be consistent with the present experimental results in consideration of this kinetic retardation of Mg_12_Zn_13_ formation. In the as-cast microstructure, it should be noted that a small amount of Mg_12_Zn_13_ was still found.

As predicted in CALPHAD calculations, alloys #1 to #8 have α-Mg as the primary phase. The solidification process of alloys #9–#12 begins differently. In these alloys, the process starts with the formation of the Mg_5_Ga_2_ intermetallic phase. The Mg_2_Zn_3_ primary solidifies in alloy #13. For alloys #14–#16, solidification initiates with the MgZn_2_ phase. Upon analyzing the XRD patterns for alloy #15, it was observed that as the Zn and Ga content increase, the Mg_7_Zn_3_ + Mg_5_Ga_2_ intermetallic transforms into Mg_2_Ga and MgZn_2_. This finding is slightly different from the calculated results, which predict the presence of (MgZn_2_ + MgGa + Mg_2_Ga) assemblage for alloys #14 and #15 and (MgZn_2_ + MgGa + MgGa_2_) for alloy #16 in the alloy’s structure. Figure 4 presents a summary of the results obtained from the CALPHAD calculations and thermal analyses. It displays the calculated liquidus projection and primary solidification areas of the Mg corner of the Mg-Zn-Ga phase diagram. The details of eutectic (E_1_, E_2_) and peritectic (P_1_, P_2_) phase transitions that shown in the liquidus projection of phase diagram (Figure 4a) presented in Table 2. As illustrated in Figure 4b, the trend of the calculated liquidus temperature for the Mg-Ga-Zn system is consistent with the experimental data. Within the same primary phase region, alloys with similar compositions exhibit both negative and positive Δ*T* (experimental-calculated) values.

The alloys with compositions #4, #8, #9, and #13 were selected for analysis in order to investigate the process of amorphization. The compositions of the #4 and #8 alloys are in close proximity to the E_2_ ternary eutectic point and the lowest liquidus temperatures of 341.5 and 328.1 °C, respectively (Figure 4a), which correspond to high GFA [63,64]. In accordance with the phase diagram, these alloys are in the area of α-Mg phase primary solidification. Alloys #9 and #13 were selected for further investigation due to their moderate freezing range (liquidus temperatures of 383.9 and 333 °C, respectively) and their positioning within the regions of primary solidification of the Mg_5_Ga_2_ and Mg_2_Zn_3_ intermetallic phases, respectively. Alloys with a composition corresponding to the areas of phase diagrams, where the primary solidification of intermetallics is observed, are preferable in terms of GFA in comparison with areas where solid solutions based on one element solidify first [63,64].

### 3.2. Results of XRD Analysis of Ribbons Made of #4 Mg_78_Zn_17_Ga_5_, #8 Mg_76_Zn_15_Ga_9_, #9 Mg_77_Zn_10_Ga_13_, and #13 Mg_69_Zn_20_Ga_11_ Alloys

Figure 5 displays the XRD spectra of samples #4 Mg_78_Zn_17_Ga_5_, #8 Mg_76_Zn_15_Ga_9_, #9 Mg_77_Zn_10_Ga_13_, and #13 Mg_69_Zn_20_Ga_11_ after melt spinning at tangential velocities of 12 and 25 m/s. Samples #4, #8 and #13 obtained at both velocities exhibit a typical amorphous structure, whereas alloy #9 after melt spinning shows clear Bragg peaks of α-Mg, Mg-Zn and Mg-Ga intermetallics. However, it should be noted that the crystallinity of #9 is higher for the alloy obtained at 12 m/s. It could be argued that the capacity to form amorphous structures increases as the tangential velocity increases. This is achieved by reducing the thickness of the ribbon and increasing the cooling rate of the air side of the ribbon. The XRD patterns of alloys #4 and #8 exhibit single Bragg peaks of Mg_2_Ga and α-Mg, while alloy #13 shows no peaks, indicating superior amorphization ability.

### 3.3. Results of DSC Analysis of Master Alloys and Ribbons Made of #4 Mg_78_Zn_17_Ga_5_, #8 Mg_76_Zn_15_Ga_9_, #9 Mg_77_Zn_10_Ga_13_ and #13 Mg_69_Zn_20_Ga_11_ Alloys

In Figure 6a, the DSC curves of alloys #4 Mg_78_Zn_17_Ga_5_, #8 Mg_76_Zn_15_Ga_9_, #9 Mg_77_Zn_10_Ga_13_, and #13 Mg_69_Zn_20_Ga_11_ are displayed. The alloys began solidifying at different temperatures, ranging from 327 °C for #8 to 376 °C for #9. Both the melting and solidification of the alloys are characterized by multiple phase transitions of primary crystals and eutectic-peritectic solidification. It was found that the solidus temperature, associated with the ternary eutectic transition, was determined to be 312–313 °C by heating curve analysis and was the same for all alloys. However, on the cooling curve, the ternary eutectic transition begins at a temperature of 294–296 °C for most of the alloys, indicating a high susceptibility to supercooling.

Figure 6b displays the DSC heating curves for the amorphous alloy ribbons of #4 Mg_78_Zn_17_Ga_5_, #8 Mg_76_Zn_15_Ga_9_, #9 Mg_77_Zn_10_Ga_13_ and #13 Mg_69_Zn_20_Ga_11_, which were obtained by spinning at a tangential velocity of 25 m/s. All alloys demonstrate the onset of crystallization from the amorphous phase; no region of supercooled liquid is observed on the DSC curves. With the exception of #9, all alloys exhibit two distinct exothermic peaks. The onset of crystallization temperature of the first exothermic peak (Tx) is 102, 115, 108 and 128 °C for #4, #8, #9 and #13 alloy ribbons. It appears that increasing the Zn and Ga content in the ribbons may result in an increase in the thermal stability of the amorphous phase. Additionally, it seems that the crystallization process is completed at a temperature above 160 °C for all ribbons.

### 3.4. Mechanical Properties of Ribbons Made of #4 Mg_78_Zn_17_Ga_5_, #8 Mg_76_Zn_15_Ga_9_, #9 Mg_77_Zn_10_Ga_13_, and #13 Mg_69_Zn_20_Ga_11_ Alloys

In Figure 7a, the microhardness of amorphous ribbons obtained from alloys #4 Mg_78_Zn_17_Ga_5_, #8 Mg_76_Zn_15_Ga_9_, #9 Mg_77_Zn_10_Ga_13_, and #13 Mg_69_Zn_20_Ga_11_ at tangential velocities of 12 and 25 m/s is shown. The results suggest that an increase in the total content of Zn and Ga corresponds to an increase in the microhardness of the ribbons. Specifically, at a velocity of 25 m/s, alloys #9 and #13 exhibit a hardness of 141 and 231 HV, respectively. However, it appears that the copper wheel rotation speed did not have a significant effect on microhardness. While it is generally accepted that hardness is proportional to strength in metallic alloys [65], the results obtained in this study do not provide clear support for this relationship. Figure 7b indicates no discernible dependence between ultimate tensile strength (UTS) and ribbon composition. Similarly, it is challenging to observe any correlation between copper wheel rotational speed and UTS. The casting defects on the ribbons have been identified as the reason for the reduction in UTS. It is worth noting that a UTS of 523.7 ± 43.9 MPa was achieved for #4 at a velocity of 25 m/s.

### 3.5. Corrosion Properties of Ribbons Made of #4 Mg_78_Zn_17_Ga_5_, #8 Mg_76_Zn_15_Ga_9_, #9 Mg_77_Zn_10_Ga_13_, and #13 Mg_69_Zn_20_Ga_11_ Alloys

Figure 8 displays the results of immersion corrosion tests for amorphous ribbons of #4 Mg_78_Zn_17_Ga_5_, #8 Mg_76_Zn_15_Ga_9_, #9 Mg_77_Zn_10_Ga_13_, and #13 Mg_69_Zn_20_Ga_11_ alloys obtained at a velocity of 12 and 25 m/s. A reduction in the hydrogen release rate was observed over time following the 50 h time point for the majority of the investigated alloy compositions. It can be postulated that the protective effect leading to a decreased corrosion rate is due to the formation of a corrosion product layer that acts as a barrier, preventing direct contact between the ribbon surface and the corrosive media. Nevertheless, in the case of ribbons with compositions #4 and #8, produced at a tangential velocity of 25 m/s, the rate of hydrogen evolution significantly rises after 130 h of immersion.

In Figure 9a, the corrosion rate (CR) of the Mg-Zn-Ga ribbons after a 192 h immersion corrosion test in Hanks’ solution at 37 °C was presented. The results suggest that there is no significant difference in CR between the alloys, as the corrosion rate falls within the range of 0.1–0.3 mm/year. However, it is challenging to determine the impact of the tangential velocity on the CR because the error bar is large. The ribbon samples #4 and #8, obtained at a tangential velocity of 25 m/s, exhibited the highest corrosion rate (CR) of approximately 0.3 mm/year. At a velocity of 12 m/s, the CR is slightly lower due to the more stable corrosion behavior associated with higher-quality ribbons. The results indicate that the #9 and #13 ribbons at 25 m/s, as well as the #8 ribbon at 12 m/s, exhibit the lowest corrosion rates with a CR close to 0.1 mm/year.

Figure 9b illustrates the pH levels of the corrosive media during the immersion corrosion tests of the Mg-Zn-Ga ribbons immersed in Hanks’ solution at 37 °C. The pH of the corrosion media was found to range from 7.6 to 8.9 for most ribbons, although the pH measurements had high uncertainty. The pH of the media during immersion of the #8 ribbon was found to be 7.6–7.7 on both velocities, which is very close to the initial pH of Hank’s solution before the corrosion tests (7.4). On the other hand, the pH of the corrosive media for the #4 ribbon was found to be 8.6–8.9 at the end of the immersion corrosion test. Thus, our results do not indicate a correlation between the CR and pH of the medium, which may differ from other corrosion test results for magnesium alloys [61,66,67].

The polarization curves for the ribbons at velocity of 12 and 25 m/s, obtained in Hanks’ solution at 37 °C, are presented in Figure 10a–d. The cathode and anode regions of curves, as well as the transition between them, are distinguished by current oscillations. This type of oscillation is documented in the literature [68,69,70,71], where it is defined as a competition between the formation of a film and its dissolution. To ensure clarity, four to five curves were selected for each regime due to high discrepancy between the curves for the same alloy and cooling rate. The curves obtained at a velocity of 12 m/s are depicted in shades of red, while those obtained at 25 m/s are depicted in shades of blue. It appears that only the #13 alloy ribbon exhibited close potentiodynamic curves at different wheel rotational speeds. In contrast, the curves for the other alloy ribbons showed a significant difference in *E_corr_* at varying copper wheel speeds. Of particular interest is the curve for the #9 ribbon obtained at 25 m/s, where the first *E_corr_* is approximately −1.5 V and the second *E_corr_* is near −0.5 V.

In Figure 11a, the electrochemical corrosion tests results for amorphous ribbons at velocities of 12 and 25 m/s are presented, showing the corrosion potential (*E_corr_*). At a tangential velocity of 12 m/s, the *E_corr_* ranges from −1.32 to −1.38 V for all ribbons. However, it is worth noting that at a velocity of 25 m/s, the *E_corr_* for #4, #8, and #9 ribbons is more positive and close to −0.5 V.

It should be noted that for ribbon samples #9 at velocities of 12 and 25 m/s, strong electrochemical heterogeneity was observed. They were also characterized by significant embrittlement during sample preparation. This indirectly indicates the presence of a crystalline phase on the surface of the ribbons or in the amorphous matrix.

Figure 11b displays the corrosion rates of the Mg-Zn-Ga amorphous ribbons, which were calculated using their corrosion current densities obtained via polarization corrosion tests. It was observed that there was no correlation between the corrosion rate and *E_corr_*. The highest corrosion rate was observed for ribbons obtained at a velocity of 25 m/s. It is evident that the corrosion resistance of the ribbon decreases with increasing alloying element content for ribbons obtained at a velocity of 12 m/s. It was observed that the ribbons with compositions #8 and #9 had a low corrosion rate of approximately 0.1–0.3 mm/year at both wheel rotational speeds. The corrosion rates obtained via immersion corrosion testing for the majority of the analyzed ribbons are lower than those obtained via polarization corrosion testing. This discrepancy can be explained by the shielding effect of the passive film of corrosion products formed on the surface of the ribbon in the case of a long time immersion corrosion test.

## 4. Discussion

In accordance with phase equilibrium calculations, the eutectic and peritectic transitions are observed in the Mg corner of the Mg-Zn-Ga phase diagram. The thermal analysis, microstructure and XRD investigation confirmed mostly the phase transitions predicted by CALPHAD calculations. In alloys #9 Mg_77_Zn_10_Ga_13_ and #13 Mg_69_Zn_20_Ga_11_, Mg_5_Ga_2_ and Mg_2_Zn_3_ were the primary crystals in the structure, respectively, and for two alloys, #4 Mg_78_Zn_17_Ga_5_ and #8 Mg_76_Zn_15_Ga_9_, α-Mg solidified primarily, and these were used for ribbons preparation. In accordance with XRD analysis, in alloys #4, #8 and #9, the formation of an amorphous–crystalline structure is observed during quenching. At the same time, the fraction of crystals in alloys #4 and #8 is extremely small, as evidenced by the low intensity of crystalline peaks compared to the amorphous diffuse maximum. In alloy #13, the formation of a completely amorphous structure is observed at all disk frequencies. The thermal analysis, DSC analysis and CAPLPHAD calculations show that alloy #9 had a higher liquidus temperature of 376 °C, and due to the solidus temperature of the investigated alloys being the same (312 °C), the freezing range of the #9 alloy is two to three times longer than that for the #4, #8 and #13 alloys. That fits well with the rule that the GFA is higher for close to eutectic alloys with a small fraction of primary crystals and short freezing range [55]. The increase in velocity from 12 to 25 m/s for alloy #9 led to a decrease in the peak height on the XRD pattern, because the crystallinity decreases as the cooling rate increases.

The heating DSC curves for the #4 Mg_78_Zn_17_Ga_5_, #8 Mg_76_Zn_15_Ga_9_, #9 Mg_77_Zn_10_Ga_13_ and #13 Mg_69_Zn_20_Ga_11_ alloys show that for all of them, crystallization peaks are observed. First of all, this confirmed the amorphous structure of alloys. At the same time, this means that when alloys are heated to temperatures higher than 100 °C, the crystallization process occurs, which can have an effect on the mechanical and corrosion behavior of the obtained ribbons. The biodegradable devices operate at 36.6 °C and must maintain the properties level up to 3–4 months. The incubation period for the crystallization of alloy #13 Mg_69_Zn_20_Ga_11_ was measured at 115, 110, and 100 °C, resulting in values of 200, 1200, and 5500 s, respectively. The dependence of the incubation period on temperature was then approximated in accordance with Equations (1) and (2):*t_o_* = *K* · exp(−*E*/*RT*)(1)
ln(*t*_o_) = ln(*K*) − *E*/*RT*(2)
where *t_o_*—incubation period, s; *E*—activation energy of crystallization under isothermal heating, J/mol; and *T*—temperature, K.

It was calculated that the incubation period for crystallization at a temperature of 36.6 °C will be 7.5 × 10^15^ s, which is more than 200 million years. In work [72], the crystallization process of the amorphous Mg_72_Zn_24_Ca_4_ alloy at the human body temperature was modeled by the Johnson–Mehl–Avrami equation. These findings confirm that the crystallization process is also a very long one with its beginning occurring only after 13 years and its end only after a thousand years. In light of the aforementioned findings, it can be posited that amorphous magnesium alloy ribbons are materials that remain stable at temperature of 36.6 °C.

The UTS of the ribbons is changed significantly for investigated alloys from 153 to 524 MPa, but this is evidence of the low quality of the ribbons due to the difference in thickness and ragged edges. For well-investigated Mg-Zn-Ca MGs, the compressive fracture strength is in the range of 500–1100 MPa [16,18,21,24,73,74], but the tensile strength and elongation do not exceed 200 MPa and 0.6%, respectively [16,75,76]. The more impressive tensile strength of ~400 MPa is obtained for Mg-Zn-Yb-Ag metallic glasses in the form of ribbons [77]. Although in this work, the Mg-Zn-Ga MGs ribbon without special preparation with significant defects was used, it showed a very high value of UTS up to 523.7 ± 43.9 MPa. This value is higher than that observed for the majority of magnesium-based MGs for which tensile strength has been measured [16,75,76,77]. An increase in the concentration of zinc in an alloy results in a corresponding enhancement in the microhardness of the alloy. The deformation of metallic glasses is initiated by the nucleation and propagation of shear bands [78]. In contrast to crystalline materials, where deformation is facilitated by dislocation sliding and the rupture of a few atomic bonds, the deformation in metallic glasses is associated with the rearrangement of a substantial number of atomic bonds. Therefore, the strength of metallic glasses is closely correlated with the elastic modulus. The dissolution of zinc in magnesium reduces the interatomic distance and increases the elastic modulus, resulting in the enhanced strength and hardness of the investigated Mg-Zn-Ga ribbons [79,80,81]. The obtained values of microhardness in the range from 141 to 231 HV are close to the range for the Mg-Zn-Ca bulk metallic glasses (210 to 300 HV) [16,75,76]. Overall, the obtained mechanical properties meet the major of requirements for biodegradable implants for orthopedic applications except for elongation at fracture [82]. To solve the problem of BMGs brittleness, the only possible way is the development of bulk metallic glass matrix composites (BMGMCs) [82,83].

The results of the immersion and polarization corrosion tests indicate that the CR of the majority of the investigated ribbon specimens falls within the range of 0.1–0.3 mm/year. Furthermore, the low pH level observed during the immersion corrosion tests supports the conclusion that the investigated ribbons exhibit high corrosion resistance.

The corrosion of Mg-based alloys is characterized by the anodic (Equation (3)) and cathodic (Equation (4)) reactions:Mg→Mg^2+^ + 2e^−^(3)
2H_2_O + 2e^−^→H_2_ + 2OH^−^(4)

As a result, on the Mg matrix, the layer of magnesium hydroxide formed due to the reaction (Equation (5)). It has been previously reported that the growth of the Mg(OH)_2_ layer is controlled by a precipitation–dissolution mechanism [84]. This is attributed to the influence of Cl^−^, which preferentially combines with Mg^2+^ to form soluble MgCl_2_, which leads to the removal of Mg(OH)_2_ segments into solution [85].
Mg^2+^ + 2OH^−^→Mg(OH)_2_(5)

In Hanks’ solution, the presence of Ca and P led to phosphate compounds’ precipitation on the surface of the hydroxide as the corrosion proceeded [45]. In accordance with reference [17], when a high Zn content is present in Mg bulk metallic glass in areas where the protective layer of Mg(OH)₂ is damaged, it is repaired by Zn(OH)₂. In the case of the investigated alloys with high Zn and Ga contents, the formation of Ga(OH)_3_ is also a possibility. This increases the protective properties of the corrosion film, because it is more stable and insoluble in aqueous solution [28]. This may be the reason for the lower corrosion rate obtained via the immersion corrosion testing of alloys #9 and #13, which have a higher Ga content.

Interesting results was obtained via a polarization corrosion test. At a velocity of 12 m/s, the *E_corr_* ranges from −1.32 to −1.38 V for all ribbons. However, at a velocity of 25 m/s, the *E_corr_* for #4, #8, and #9 ribbons is more positive and close to −0.5 V. The polarization curve for the #9 ribbon obtained at 25 m/s produced two *E_corr_* values: the first *E_corr_* is approximately –1.5 V, and the second *E_corr_* is near −0.5 V. At the same time, the ribbon with composition #9 had a low corrosion rate of approximately 0.2–0.3 mm/year at both rotational speeds despite the observation of two *E_corr_* values (−1.5 V and −0.5 V). As usual, the amorphous alloys corrosion potential after crystallization was more negative and changed by 0.1–0.4 V [19,20,29,30,86,87,88,89]. This indicates that for ribbon #9, which comprises both amorphous and crystalline components, the *E_corr_* −0.5 V correlates with the amorphous portion, whereas the *E_corr_* −1.5 V correlates with the crystalline portion.

However, Zai et al. show that Ga addition led to the positive shift of the corrosion potential of Mg-Zn-Ca amorphous alloys up to +0.1 V with the addition of 1 at.% Ga. In comparison with crystalline magnesium alloys, the corrosion potential of magnesium metallic glasses can change in a wide range due to the high content of alloying elements. For example, for Mg_62_Li_4_Zn_30_Ca_4_, Mg_70_Al_15_Ga_15_ and Mg_75_Cu_13_Y_7_Zn_5_ BMGs, the *E_corr_* value was −0.2V, −1.8 and −1.03, respectively [27,30,53].

The formation of the ribbon at a velocity of 25 m/s results in the development of a predominantly amorphous structure, as evidenced by XRD analysis. The formation of an amorphous–crystalline structure was observed in ribbons obtained at a low rotational frequency (12 m/s). Therefore, the shift of *E_corr_* to more noble values at 25 m/s is indicative of the formation of a predominantly amorphous structure devoid of secondary phases and defects associated with a crystalline structure, such as grain boundaries and dislocations. With regard to the ribbon of alloy #13 Mg_69_Zn_20_Ga_11_, obtained at both rotational frequencies, similar potentiodynamic curves were observed. This is attributable to the fact that the alloys obtained at different rotational speeds possess the same structure. However, in comparison with other predominantly amorphous alloys (specifically, #4 Mg_78_Zn_17_Ga_5_, #8 Mg_76_Zn_15_Ga_9_, and #9 Mg_77_Zn_10_Ga_13_), which were obtained at a wheel rotational frequency of 25 m/s, a negative shift in *E_corr_* was observed. This behavior may be attributed to the highest Zn content in comparison with other alloys. An increase in Zn content results in an increase in Mg-Zn atomic pairs within the structure, which may be a contributing factor to the observed shift in *E_corr_*. For instance, the corrosion potential of Mg-Zn-Ca-based amorphous alloys that are rich in Zn is highly negative, ranging from −1.1 to −1.35 V, as evidenced in previous studies [16,26,73,77,90,91,92].

Typically, amorphous structures exhibit superior corrosion resistance compared to their crystalline counterparts. This is attributed to the uniform corrosion observed in amorphous magnesium alloys, which is a consequence of the amorphous phase’s chemical homogeneity [90]. In instances of partial crystallization, microgalvanic corrosion can also be observed, resulting in an increased corrosion rate. However, the results do not indicate a significant increase in corrosion rate for ribbon #9, which exhibits the highest crystallinity among the alloys, which is in accordance with the XRD findings. Wang et al. show that the partially crystallized Mg-Zn-Ca alloy showed the best corrosion performance in terms of pitting resistance, whereas the fully amorphous alloy showed the highest pitting susceptibility [88]. The filiform and pitting corrosion can be major problems due to the amorphous structure [93].

It is difficult to strongly establish the influence of alloy composition and the rotational frequency of copper wheel on the corrosion rate due to the effect of ribbon quality on the corrosion test results. As usual, corrosion tests (both electrochemical and immersion) administered on bulk cylindrical BMGs samples indicated higher quality [17,73]. It is possible that further investigation of the bulk Mg-Zn-Ga BMGs samples sheds light on the influence of these parameters on the CR. However, the obtained corrosion rate range for Mg-Zn-Ga amorphous alloys is close to the corrosion rate of Mg-Zn-Ca based amorphous alloys (0.05–0.4) mm/year [16,18,23,24,74,90,92]. Also, it can be seen that all of the analyzed samples in accordance with the immersion corrosion test meet the requirements for biodegradable implants (corrosion rate < 0.5 mm/year [94]).

## 5. Conclusions

The liquidus projection of the Mg-Zn-Ga phase diagram and solidification pathway of Mg-Zn-Ga alloys were calculated via the CALPHAD method and confirmed by thermal analysis and microstructure observations. The ribbons of four alloys, #4 Mg_78_Zn_17_Ga_5_, #8 Mg_76_Zn_15_Ga_9_, #9 Mg_77_Zn_10_Ga_13_, and #13 Mg_69_Zn_20_Ga_11_, were produced by melt spinning. The following results are obtained:
(i)The XRD and DSC analysis confirmed that the ribbon of the #13 Mg_69_Zn_20_Ga_11_ alloy has a fully amorphous structure, and the #4 Mg_78_Zn_17_Ga_5_, #8 Mg_76_Zn_15_Ga_9_ and #9 Mg_77_Zn_10_Ga_13_ ribbons are partially crystalline. The onset of the crystallization temperature of the first exothermic peak (Tx) is in the range of 102–128 °C, and increasing the Zn and Ga content in the ribbons may result in a higher Tx.(ii)The ribbons’ microhardness was 141 and 231 HV and increased with the increase in the total content of Zn and Ga in alloys, but the rotational speed of the melt spinner wheel had no effect on the microhardness. The tensile properties of the ribbons have a large influence on the ribbon quality, and the maximal UTS of 524 MPa was observed for #4 Mg_78_Zn_17_Ga_5_ ribbon.(iii)The results of an immersion corrosion test show that the CR of the investigated ribbons is in the range of 0.1–0.3 mm/year. This is in accordance with requirements to biodegradable materials, making the Mg-Zn-Ga ribbons a candidate material for biodegradable applications. The potentiodynamic corrosion test shows that ribbon #9 Mg_77_Zn_10_Ga_13_ has both amorphous and crystalline parts because there were two corrosion potentials, −0.5 and −1.5 V, observed in the polarization curve.

The objective of the subsequent research phase is to obtain the bulk Mg-Zn-Ga amorphous specimens. Following this, an in vitro analysis of the Mg-Zn-Ga alloys biocompatibility will be conducted, specifically investigating their cytotoxicity on cell cultures and in vivo investigation on animals. In addition, the potential for coating the bulk amorphous specimens will be explored.

## Figures and Tables

**Figure 1 jfb-15-00275-f001:**
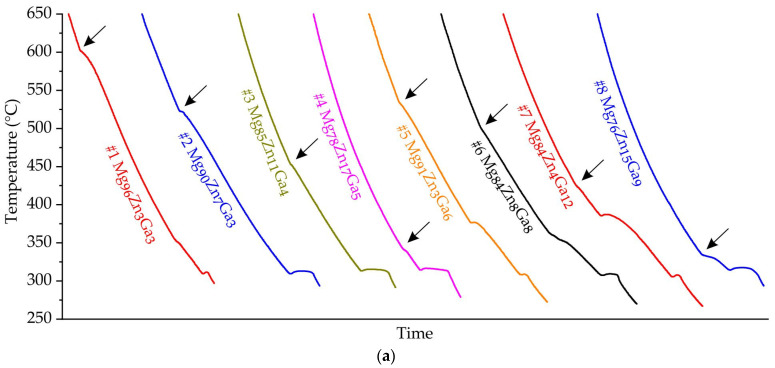
The cooling curves of investigated Mg-Zn-Ga alloys: (**a**) #1–#8, (**b**) #9–#16. Liquidus temperatures shown by arrows.

**Figure 2 jfb-15-00275-f002:**
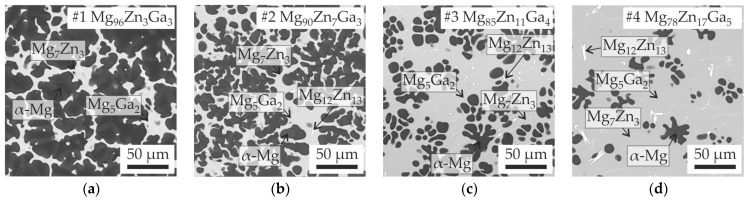
Microstructure of (**a**) #1 Mg_96_Zn_3_Ga_3_, (**b**) #2 Mg_90_Zn_7_Ga_3_, (**c**) #3 Mg_85_Zn_11_Ga_4_, (**d**) #4 Mg_78_Zn_17_Ga_5_, (**e**) #5 Mg_91_Zn_3_Ga_6_, (**f**) #6 Mg_84_Zn_8_Ga_8_, (**g**) #7 Mg_84_Zn_4_Ga_12_, (**h**) #8 Mg_76_Zn_15_Ga_9_, (**i**) #9 Mg_77_Zn_10_Ga_13_, (**j**) #10 Mg_75_Zn_6_Ga_19_, (**k**) #11 Mg_67_Zn_11_Ga_22_, (**l**) #12 Mg_66_Zn_17_Ga_17_, (**m**) #13 Mg_69_Zn_20_Ga_11_, (**n**) #14 Mg_51_Zn_25_Ga_24_, (**o**) #15 Mg_49_Zn_20_Ga_31_, (**p**) #16 Mg_34_Zn_31_Ga_35_ alloys.

**Figure 3 jfb-15-00275-f003:**
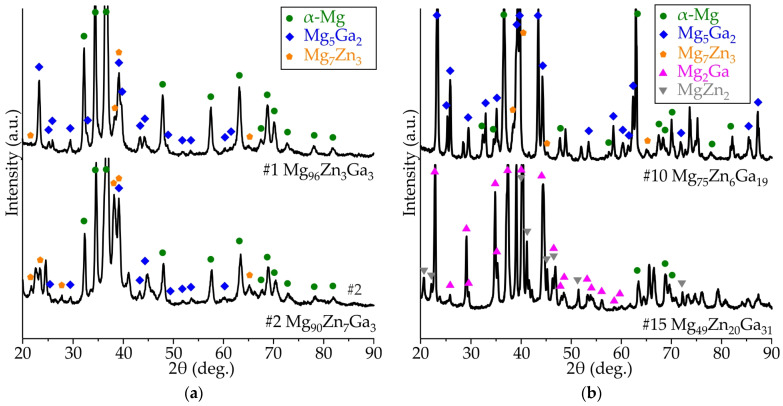
The XRD patterns of Mg-Zn-Ga alloys: (**a**) #1 Mg_96_Zn_3_Ga_3_ and #2 Mg_90_Zn_7_Ga_3_, (**b**) #10 Mg_75_Zn_6_Ga_19_ and #15 Mg_49_Zn_20_Ga_31_ alloys.

**Figure 4 jfb-15-00275-f004:**
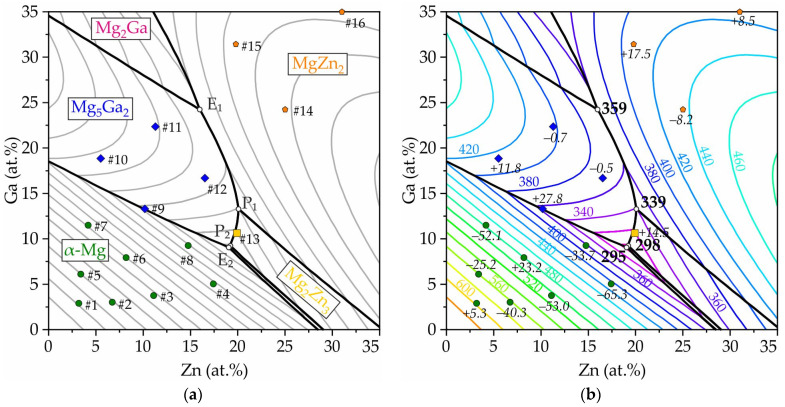
Calculated via CALPHAD method liquidus projection of the Mg-Zn-Ga phase diagram: (**a**) with points of the prepared alloys and the areas of primary solidification, (**b**) with the isotherms and points showing a difference between experimental and calculated liquidus temperature.

**Figure 5 jfb-15-00275-f005:**
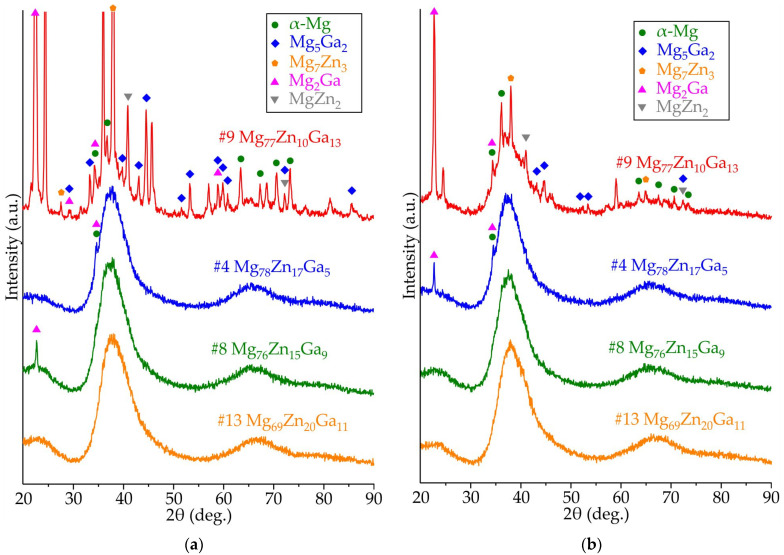
XRD spectra of the samples #4 Mg_78_Zn_17_Ga_5_, #8 Mg_76_Zn_15_Ga_9_, #9 Mg_77_Zn_10_Ga_13_, and #13 Mg_69_Zn_20_Ga_11_ after melt spinning at tangential velocity of (**a**) 12 m/s and (**b**) 25 m/s.

**Figure 6 jfb-15-00275-f006:**
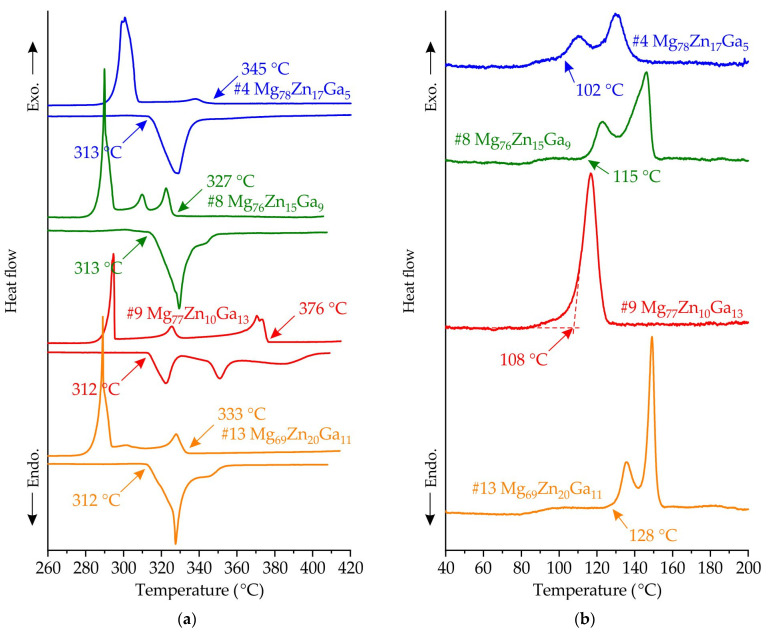
The DSC curves of alloys #4 Mg_78_Zn_17_Ga_5_, #8 Mg_76_Zn_15_Ga_9_, #9 Mg_77_Zn_10_Ga_13_ and #13 Mg_69_Zn_20_Ga_11_ in (**a**) as-cast (crystalline) sate, and (**b**) in the amorphous sate (ribbons) obtained by spinning at velocity of 25 m/s. Liquidus and solidus temperatures shown by arrows.

**Figure 7 jfb-15-00275-f007:**
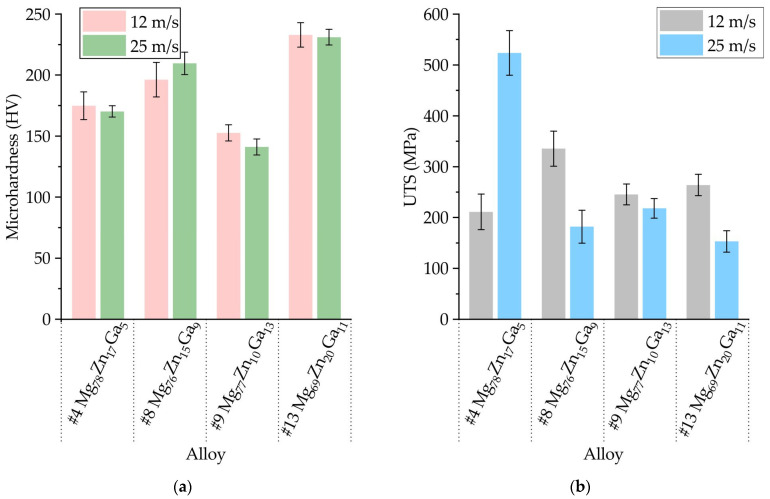
The (**a**) microhardness and (**b**) UTS of, #4 Mg_78_Zn_17_Ga_5_, #8 Mg_76_Zn_15_Ga_9_, #9 Mg_77_Zn_10_Ga_13_ and #13 Mg_69_Zn_20_Ga_11_ ribbons obtained by spinning at a tangential velocity of 12 and 25 m/s.

**Figure 8 jfb-15-00275-f008:**
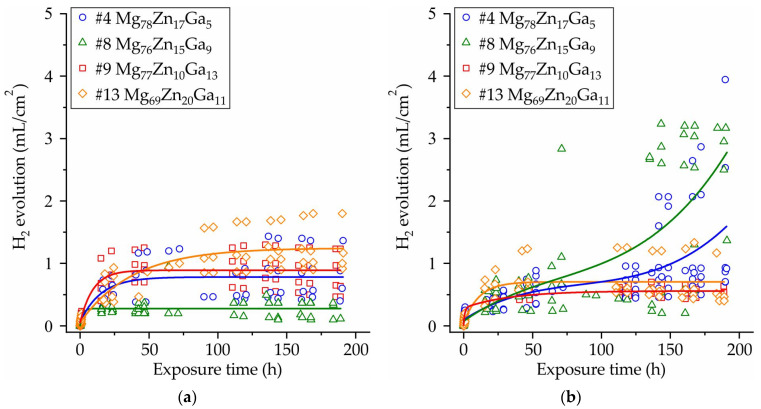
Hydrogen evolution during 192 h of immersion in Hanks’ solution at 37 °C of #4 Mg_78_Zn_17_Ga_5_, #8 Mg_76_Zn_15_Ga_9_, #9 Mg_77_Zn_10_Ga_13_ and #13 Mg_69_Zn_20_Ga_11_ ribbons obtained by spinning at velocity of (**a**) 12 m/s and (**b**) 25 m/s.

**Figure 9 jfb-15-00275-f009:**
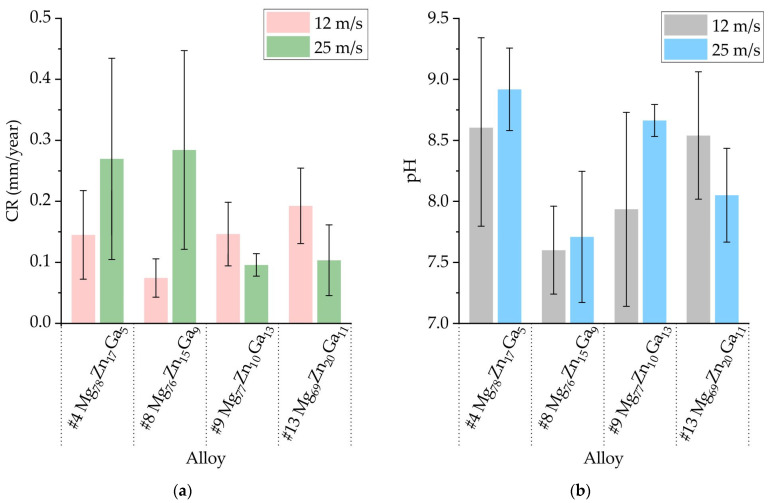
The (**a**) corrosion rate (CR) obtained from the immersion tests and (**b**) pH of the corrosion media for amorphous ribbons with composition of #4 Mg_78_Zn_17_Ga_5_, #8 Mg_76_Zn_15_Ga_9_, #9 Mg_77_Zn_10_Ga_13_ and #13 Mg_69_Zn_20_Ga_11_ obtained by spinning at velocity of 12 and 25 m/s.

**Figure 10 jfb-15-00275-f010:**
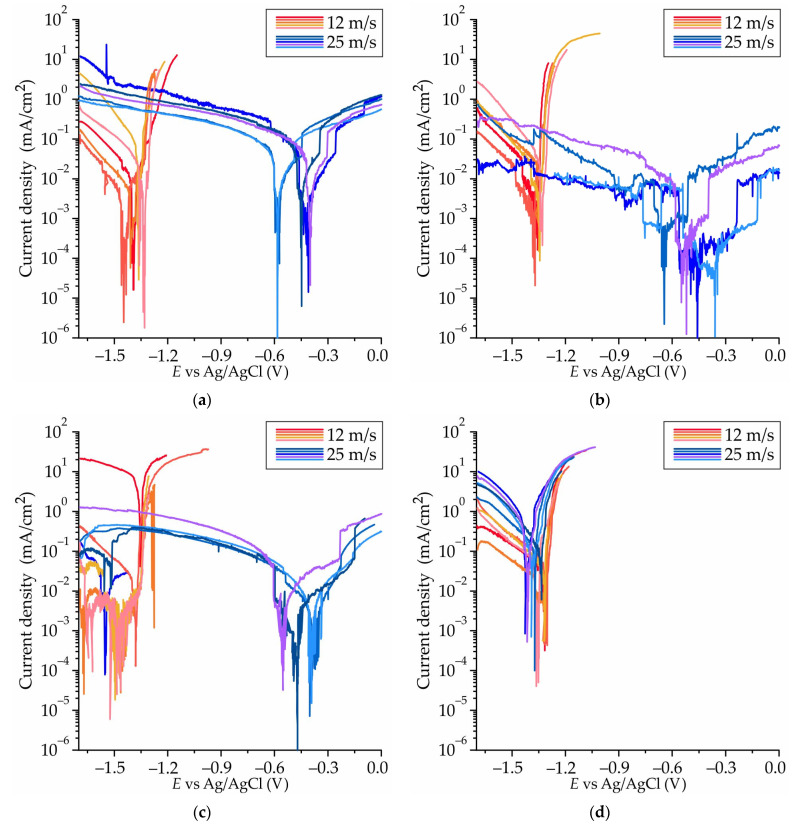
Polarization curves obtained in Hanks’ solution at 37 °C for the (**a**) #4 Mg_78_Zn_17_Ga_5_, (**b**) #8 Mg_76_Zn_15_Ga_9_, (**c**) #9 Mg_77_Zn_10_Ga_13_, and (**d**) #13 Mg_69_Zn_20_Ga_11_ amorphous ribbons obtained by spinning at velocities of 12 and 25 m/s.

**Figure 11 jfb-15-00275-f011:**
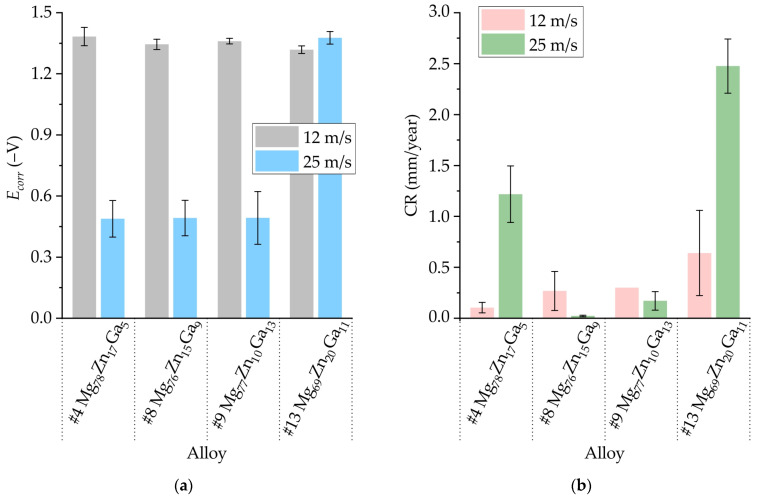
The (**a**) corrosion potential (*E_corr_*) and (**b**) corrosion rate (CR) obtained via electrochemical corrosion tests for #4 Mg_78_Zn_17_Ga_5_, #8 Mg_76_Zn_15_Ga_9_, #9 Mg_77_Zn_10_Ga_13_ and #13 Mg_69_Zn_20_Ga_11_ amorphous ribbons obtained by spinning at velocities of 12 and 25 m/s.

**Table 1 jfb-15-00275-t001:** Elemental compositions of the prepared alloys, liquidus temperatures and solidification pathways.

Alloy	Element Content (at.%)			Liquidus Temperature (°C)			Calculated Solidus Temperature (°C)	Equilibrium Solidification Pathway
	Mg	Zn	Ga	Experimental	Calculated	Error		
#1 Mg_96_Zn_3_Ga_3_	Bal.	3.2	2.9	602.8	597.5	+5.3	295	L→α-Mg; L→ α-Mg + Mg_12_Zn_13_; L→α-Mg + Mg_12_Zn_13_ + Mg_5_Ga_2_
#2 Mg_90_Zn_7_Ga_3_	Bal.	6.7	3.0	522.4	562.7	–40.3	295	
#3 Mg_85_Zn_11_Ga_4_	Bal.	11.1	3.8	452.4	505.4	–48.0	295	
#4 Mg_78_Zn_17_Ga_5_	Bal.	17.4	5.0	341.5	406.8	–65.3	295	
#5 Mg_91_Zn_3_Ga_6_	Bal.	3.4	6.1	535.3	560.6	–25.2	295	L→α-Mg; L→α-Mg + Mg_5_Ga_2_; L→α-Mg + Mg_12_Zn_13_ + Mg_5_Ga_2_
#6 Mg_84_Zn_8_Ga_8_	Bal.	8.2	8.0	502.6	479.4	+23.2	295	
#7 Mg_84_Zn_4_Ga_12_	Bal.	4.2	11.5	424.8	476.9	–52.1	295	
#8 Mg_76_Zn_15_Ga_9_	Bal.	14.8	9.3	328.1	361.8	–33.7	295	
#9 Mg_77_Zn_10_Ga_13_	Bal.	10.2	13.3	383.9	356.1	+27.8	295	L→Mg_5_Ga_2_; L→α-Mg + Mg_5_Ga_2_; L→α-Mg + Mg_12_Zn_13_ + Mg_5_Ga_2_
#10 Mg_75_Zn_6_Ga_19_	Bal.	5.5	18.9	425.3	413.5	+11.8	295	
#11 Mg_67_Zn_11_Ga_22_	Bal.	11.3	22.4	401.8	402.5	–0.7	298	L→Mg_5_Ga_2_; L→Mg_5_Ga_2_ + MgZn_2_; L + MgZn_2_→ Mg_5_Ga_2_ + Mg_2_Zn_3_; L→Mg_5_Ga_2_ + Mg_2_Zn_3_ + Mg_12_Zn_13_
#12 Mg_66_Zn_17_Ga_17_	Bal.	16.5	16.7	368.2	368.7	–0.5	295	L→Mg_5_Ga_2_; L→Mg_5_Ga_2_ + MgZn_2_; L + MgZn_2_→Mg_5_Ga_2_ + Mg_2_Zn_3_; L + Mg_2_Zn_3_→Mg_5_Ga_2_ + Mg_12_Zn_13_; L→α-Mg + Mg_12_Zn_13_ + Mg_5_Ga_2_
#13 Mg_69_Zn_20_Ga_11_	Bal.	19.9	10.6	333	318.5	–14.8	295	L→Mg_2_Zn_3_; L→Mg_2_Zn_3_ + Mg_5_Ga_2_; L + Mg_2_Zn_3_→Mg_5_Ga_2_ + Mg_12_Zn_13_; L→α-Mg + Mg_12_Zn_13_ + Mg_5_Ga_2_
#14 Mg_51_Zn_25_Ga_24_	Bal.	25.0	24.2	428.2	436.4	–8.2	317	L→MgZn_2_; L→MgZn_2_ + Mg_2_Ga; L→MgZn_2_ + Mg_2_Ga + MgGa
#15 Mg_49_Zn_20_Ga_31_	Bal.	19.8	31.4	425.0	407.5	+17.5	317	
#16 Mg_34_Zn_31_Ga_35_	Bal.	31.1	35.0	421.4	412.9	+8.5	259	L→MgZn_2_; L→MgZn_2_ + MgGa_2_; L→MgZn_2_ + MgGa + MgGa_2_

**Table 2 jfb-15-00275-t002:** The nonvariant phase transitions presented in liquidus projection of the Mg-Zn-Ga phase diagram (Figure 4).

Symbol	Temperature (°C)	Transition
E_1_	359	L→MgZn_2_ + Mg_2_Ga + Mg_5_Ga_2_
P_1_	339	L + MgZn_2_→Mg_2_Zn_3_ + Mg_5_Ga_2_
P_2_	298	L + Mg_2_Zn_3_→Mg_12_Zn_13_ + Mg_5_Ga_2_
E_2_	295	L→α-Mg + Mg_12_Zn_13_ + Mg_5_Ga_2_

## Data Availability

The original contributions presented in the study are included in the article, further inquiries can be directed to the corresponding author.

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
