# Peer review of "Investigation of Mechanical and Corrosion Properties of New Mg-Zn-Ga Amorphous Alloys for Biomedical Applications"

_jfb, 2024, doi:10.3390/jfb15090275_

Round 1

Reviewer 1 Report

Comments and Suggestions for Authors

In the manuscript there are interesting results concerning the mechanical and corrosion properties of MgZnGa and MgZnCa bulk metallic glasses. The influence of a spinning velocity on the corrosion rate has been investigated. However, the manuscript needs significant improvement, because it lacks important information and contains experimental errors.

1. Fig. 10 (b) blue polarization curves are not correct. There is no transition from cathodic to anodic branch. The curves are noisy. The shape of the LSV polarization curves is not correct. The experiment should be repeated.

2. The authors should explain why the polarization curves are shifted towards the anodic direction for amorphous samples made at spinning velocity of 25 m/s.

3. 3. The Mg77Zn10Ga13 sample is partially crystalline. The authors should explain how the presence of crystalline phases affects the corrosion rate of the tested material.

4. The authors did not observe the surface of the samples after corrosion tests. The chemical composition of the corrosion products is unknown.

5. What type of corrosion is experienced by amorphous samples (pitting corrosion?, galvanic, uniform...)

6. Authors should describe cathodic and anodic reactions.

7. Authors should describe the corrosion mechanism of MgZnGa alloys. What role does gallium play in corrosion?

8. In order to understand the corrosion mechanism of Mg alloys in the Hank’s solution the electrochemical impedance measurements should be performed.

Author Response

In the manuscript there are interesting results concerning the mechanical and corrosion properties of MgZnGa and MgZnCa bulk metallic glasses. The influence of a spinning velocity on the corrosion rate has been investigated. However, the manuscript needs significant improvement, because it lacks important information and contains experimental errors.

Comment 1: Fig. 10 (b) blue polarization curves are not correct. There is no transition from cathodic to anodic branch. The curves are noisy. The shape of the LSV polarization curves is not correct. The experiment should be repeated.

Response 1: The polarization curves presented in Fig. 10b were replicated numerous times, and all of these curves exhibited considerable noise. Other curves also displayed noise, albeit to a lesser extent. The cathode and anode regions, as well as the transition between them, exhibit current oscillations. This phenomenon, which has been described in the literature, can be understood as a competition between the formation of a film and its dissolution. This information has been added to the article text. To enhance the quality of the curves, the curves in Fig. 10b were subjected to noise reduction.

Comment 2: The authors should explain why the polarization curves are shifted towards the anodic direction for amorphous samples made at spinning velocity of 25 m/s.

Response 2: The corrosion behavior is dependent on phase composition and structure. Because of high velocity during formation of the ribbon (25 m/s) the mostly amorphous structure is obtained. Usually the amorphous structure have the higher corrosion resistance in comparison with crystalline counterpart. In ribbons obtained at low rotational frequency (12 m/s) the formation of amorphous-crystalline structure is observed. Thus the shift of Ecorr to the more noble values at 25 m/s is correspond to the formation of mostly amorphous structure without second phase and defects of crystalline structure (grain boundaries, dislocations, etc.). As for #13 Mg69Zn20Ga11 obtained in both rotational frequencies the similar potentiodynamic curves is observed. This is connected with the same structure of the alloys obtained at different rotational speed. However, in comparison with other mostly amorphous alloys (  #4 Mg78Zn17Ga5,  #8 Mg76Zn15Ga9, #9 Mg77Zn10Ga13) obtained at wheel rotational frequency of 25 m/s the negative shift of Ecorr is observed. This behavior could be explained by the highest Zn content in comparison with other alloys. The increase of Zn content lead to increasing Mg-Zn atomic pairs in structure. This is a possible reason of the Ecorr shift. However the dependance of Ecorr on amorphous ribbon composition is need further investigation. This information added to the article.

Comment 3: The Mg77Zn10Ga13 sample is partially crystalline. The authors should explain how the presence of crystalline phases affects the corrosion rate of the tested material.

Response 3: Wang et al. using the electrochemical impedance spectroscopy (EIS) and potentiodynamic polarization show that in amorphous magnesium alloys uniform corrosion is observed because the amorphous phase has good chemical homogeneity and excellent corrosion resistance [30]. In the case of partial crystallization the microgalvanic corrosion is also can be observed and corrosion rate must increase. But our results don’t show any notable increase of corrosion rate for ribbon #9 with highest crystallinity among other alloys in accordance with XRD results. Wang et al. show that partially crystallized Mg-Zn-Ca alloy showed the best corrosion performance in terms of pitting resistance, whereas the fully amorphous alloy showed the highest pitting susceptibility [88]. The filiform and pitting corrosion can be major problems due to the amorphous structure [93]. The following information was added to the article.

Comment 4: The authors did not observe the surface of the samples after corrosion tests. The chemical composition of the corrosion products is unknown.

Response 4: At this juncture, the article has already exceeded 10,000 words, and the incorporation of the corrosion products composition result in an unwarranted increase in length. This work represents the inaugural effort to ascertain and examine the characteristics of amorphous alloys within the Mg-Zn-Ga system, with a primary emphasis on the phase diagram investigation. The corrosion and mechanical properties are employed exclusively for the assessment of the potential for biomedical applications. Additionally, numerous studies have examined the corrosion products that form during the corrosion of Mg alloys in Hanks solution, particularly in Mg-Zn-Ga alloys (https://doi.org/10.1016/j.jma.2020.02.009). The corrosion products were predominantly Mg(OH₂) and Mg,Ca phosphate. In subsequent research on bulk specimens, we will analyze the composition of corrosion products.

Comment 5: What type of corrosion is experienced by amorphous samples (pitting corrosion?, galvanic, uniform...)

Response 5: Wang et al. using the electrochemical impedance spectroscopy (EIS) and potentiodynamic polarization show that in amorphous magnesium alloys uniform corrosion is observed because the amorphous phase has good chemical homogeneity and excellent corrosion resistance [30]. In the case of partial crystallization the microgalvanic corrosion is also can be observed and corrosion rate must increase. But our results don’t show any notable increase of corrosion rate and the corrosion rate for ribbon #9 with highest сrystallinity. Wang et al. show that partially crystallized Mg-Zn-Ca alloy showed the best corrosion performance in terms of pitting resistance, whereas the fully amorphous alloy showed the highest pitting susceptibility [88]. The filiform and pitting corrosion can be major problems due to the amorphous structure [93].

Comment 6: Authors should describe cathodic and anodic reactions.

Response 6: The following part was added to discussion paper:

The corrosion of Mg-based alloys is characterized by the anodic (3) and cathodic (4) reactions:

Mg → Mg2+ + 2e−        (3)

2H2O + 2e− → H2 + 2OH−       (4)

As a result on the Mg matrix the layer of magnesium hydroxide formed on the Mg matrix due to reaction (5). It has been previously reported that the growth of the Mg(OH)2 layer is controlled by a precipitation-dissolution mechanism [84]. This is attributed to the influence of Cl−, which preferentially combines with Mg2+ to form sol-uble MgCl2, which leads to the removal of Mg(OH)2 segments into solution [85].

Mg2+ + 2OH− → Mg(OH)2       (5)

In Hanks solution the Ca and P presence leading to phosphate compounds precip-itation on the surface of the hydroxide as the corrosion proceeded [45]. In accordance with reference [17], when a high Zn content is present in Mg bulk metallic glass in ar-eas where the protective layer of Mg(OH)₂ is damaged, it is repaired by Zn(OH)₂. In the case of the investigated alloys with high Zn and Ga contents, the formation of Ga(OH)3 is also a possibility. This increases the protective properties of the corrosion film, be-cause it is more stable and insoluble in aqueous solution [28]. This may be the reason for the lower corrosion rate obtained via immersion corrosion testing of alloys #9 and #13, which have a higher Ga content.   

Comment 7: Authors should describe the corrosion mechanism of MgZnGa alloys. What role does gallium play in corrosion?

Response 7: In accordance with reference [16], when a high Zn content is present in Mg bulk metallic glass in areas where the protective layer of Mg(OH)₂ is damaged, it is repaired by Zn(OH)₂. In the case of the investigated alloy, the formation of Ga(OH)3 also increases the protective properties of the corrosion film. This is the reason for the lower corrosion rate obtained via immersion corrosion testing of alloys #9 and #13, which have a higher Ga content. 

Comment 8: In order to understand the corrosion mechanism of Mg alloys in the Hank’s solution the electrochemical impedance measurements should be performed.

Response 8: At this point in time, the article has already exceeded 10,000 words, and the addition of the EIS results would result in an unjustified increase in length. This work represents the inaugural effort to ascertain and examine the characteristics of amorphous alloys within the Mg-Zn-Ga system, with a primary emphasis on the phase diagram investigation. The corrosion and mechanical properties are utilized solely for the evaluation of the potential for biomedical applications. It is conceivable that in subsequent research involving bulk amorphous specimens, the EIS analysis of Mg-Zn-Ga alloys will be conducted and presented.

Reviewer 2 Report

Comments and Suggestions for Authors

Dear authors,

please find below some comments about your submitted manuscript.

- Regarding the introduction, there is little to none mention of any side effects or adverse reactions to use of Mg-rich implants.

-More information should be added in the methods' section, to adequately describe the equipment used, the conditions of the experiments, etc.

- In the results' section, please provide additional explanation on why you focused on samples 4,8,9, and 13 among the overall 16 samples.

- The conclusion section could be decreased in length, but please provide a plan for the future biocompability tests, e.g. cell cultures or animal studies will be the next step.

Comments on the Quality of English Language

Reviewing from Native English speaker is advised.

Author Response

Comment 1: Regarding the introduction, there is little to none mention of any side effects or adverse reactions to use of Mg-rich implants.

Response 1: This information was added.” Moreover, the release of gaseous hydrogen during the corrosion process can result in the formation of gas gangrene, which may accelerate the degradation of implant integrity and impede the proper connectivity of osteocytes to the implant surface. This could potentially impact bone tissue healing [2, 12]. It is well established that magnesium (Mg) is an essential element for human metabolism. However, an overdose can occur due to the extremely high rate of Mg ions release, which can lead to respiratory distress, hypotension, muscular paralysis, and potentially cardiac arrest [84].”

Comment 2: More information should be added in the methods' section, to adequately describe the equipment used, the conditions of the experiments, etc.

Response 2: The data regarding the apparatus and circumstances of the experiments were augmented.

Comment 3: In the results' section, please provide additional explanation on why you focused on samples 4,8,9, and 13 among the overall 16 samples.

Response 3: This part was rewritten: “The alloys with compositions #4, #8, #9, and #13 were selected for analysis in order to investigate the process of amorphization. The compositions of the #4 and #8 alloys are in close proximity to the E2 ternary eutectic point and the lowest liquidus temper-atures of 341.5 and 328.1 °C, respectively (Fig. 4a), which correspond to high GFA [85, 86]. The #9 and #13 alloys were selected for further investigation due to their moderate freezing range (liquidus temperatures of 383.9 and 333 °C, respectively) and primary solidification of the Mg5Ga2 and Mg2Zn3 intermetallic phases. Alloys with intermetallics primary solidification are preferable in terms of GFA in comparison with solid solutions based on one element [85, 86].”

Comment 4: The conclusion section could be decreased in length, but please provide a plan for the future biocompability tests, e.g. cell cultures or animal studies will be the next step.

Response 4: The first paragraph of the conclusions was shortened. In the end of conclusions, the additional information about further biocompatibility investigations was added: “The objective of the subsequent research phase is to obtain the bulk Mg–Zn–Ga amorphous specimens. Following this, an in vitro analysis of the Mg–Zn–Ga alloys biocompatibility will be conducted, specifically investigating their cytotoxicity on cell cultures and in vivo investigation on animals. In addition, the potential for coating the bulk amorphous specimens will be explored.”

Round 2

Reviewer 1 Report

Comments and Suggestions for Authors

The manuscript was significantyly improved.

I suggest to pulish it i JFB.

Reviewer 2 Report

Comments and Suggestions for Authors

Dear authors,

thank you for addressing the comments during the previous round of review.